# Synchronous/Metachronous Multiple Primary Malignancies: Review of Associated Risk Factors

**DOI:** 10.3390/diagnostics12081940

**Published:** 2022-08-11

**Authors:** Szu-Ying Pan, Chi-Ping Huang, Wen-Chi Chen

**Affiliations:** 1Department of Urology, China Medical University Hospital, Taichung 404332, Taiwan; 2School of Medicine, College of Chinese Medicine, China Medical University, Taichung 404332, Taiwan; 3Graduate Institute of Integrated Medicine, College of Chinese Medicine, China Medical University, Taichung 404332, Taiwan

**Keywords:** multiple primary malignancy, secondary primary malignancy, genitourinary, malignancies, cancer, metachronous, synchronous

## Abstract

The incidence of secondary primary malignancy (SPM) has been reported to range from 1.33% to 5.8%, according to the location of the primary cancer and the follow-up duration. The highest occurrence rate of SPM, of 36.6% within 6 months, has been reported in lung cancer. Genitourinary malignancies were reported to be the third-most-common SPM in several reports. However, the incidence of genitourinary malignancy as the first primary cancer associated with SPM has not been reported. Several risk factors are related to the occurrence of SPM, including viral infection chemotherapy, radiation, genetics, smoking, betel quid chewing, and environmental factors. An early survey for SPM is indicated in first primary malignancy patients with these associated factors. In this study, we summarize several risk factors related to the occurrence of SPMs and preventive tests, which may help in their early detection and, consequently, better survival.

## 1. Introduction

Multiple primary malignancies are not uncommon in cancer patients, and they may involve risk factors such as genetics, viral infection, smoking, betel quid chewing, and environmental or treatment-related factors. The frequency of multiple primary malignancies in the same or different organ systems ranges from 2% to 17% [1,2,3,4]. The occurrence of multiple malignancies may be synchronous in the short-term or the long-term during follow-up. Therefore, early detection, follow-up survey, and risk reduction in second/third malignancies is very important in the clinical management after primary cancer treatment.

The definition of synchronous, multiple primary malignancies differs between studies. MPM was first described in 1869, and then Warren and Gates extended the criteria to define synchronous tumors in 1932 [5]. These two tumors may be diagnosed at the same time (synchronous), or the second tumor may be detected 6 months afterwards (metachronous). A subsequent definition was provided by the Surveillance Epidemiology and End Results (SEER) project [6], which defined two or more primary malignancies diagnosed within a 2-month period as synchronous multiple tumors. In comparison, the International Association of Cancer Registries and International Agency for Research on Cancer (IACR/IARC) defined synchronous multiple primary malignancies as two or more primary malignancies diagnosed within a 6-month period [2]. Beyond this definition, more than two primary malignancies occurring at different times was defined as metachronous multiple primaries. However, the SEER definition seems to be too strict, as this criterion may miss some patients with a first diagnosis of a primary tumor, which was not surveyed within a two-month interval. The definition of multiple primary sites of the same organ also differs. SEER defines this as a single tumor. However, this may be more complicated when facing multiple sites of urothelial cell carcinomas in the urinary tract, as it is more likely that tumor cells from the upper urinary tract down-migrate to the lower urinary tract. If a tumor in the lower tract occurs 6 months after the upper tract tumor, it is unlikely to involve metachronous multiple primaries. This review discusses the incidence and risk factors for multiple primary malignancies with a special focus on genitourinary tract primary cancers with synchronous or metachronous primary cancers. Few studies have analyzed genitourinary system malignancies associated with synchronous primary malignancy.

## 2. Incidence of Multiple Primary Malignancies

Tanjak et al. retrospectively investigated 109,054 adult patients with a first solid cancer and followed them for the occurrence of a second primary cancer over a 25-year period [7]. They defined a 2-month period between the first and second primary cancers as synchronous multiple primary cancers, and cancers occurring after 2 months as metachronous multiple primary cancers. Their results showed that 1785 (1.63%) patients developed multiple primary cancers, and that most (70.87%) occurred after 2 months. The first cancers were breast, skin, colorectal, lung, head and neck, liver, prostate, thyroid, and female non-uterine genital cancers. Head and neck cancers had the highest metachronous association with second esophageal cancers (relative risk (RR), 25.06; 95% CI, 13.41–50.77). Prostate cancer and second colorectal cancer had the second-highest metachronous association (RR, 2.00; 95% CI, 1.25–3.05). There was also a strong synchronous association between uterine and ovarian cancers (RR, 27.77; 95% CI, 17.97–43.63). The shortest median time to develop a second cancer was 55 days in the patients with uterine cancer.

In Taiwan, Chen et al. analyzed 23,639 patients with nasopharyngeal carcinoma from a cancer registration database in Taiwan from 1 January 1979 to 31 December 2003 [8], of whom 712 (3.01%) developed secondary primary malignancy (SPM) with an average time interval of 5.33 years. A higher incidence of second cancers was found in patients with oral cancer, a major salivary gland tumor, sarcoma, and skin cancer. Prostate cancer was less frequently seen in this analysis and was reported as having a low risk of a second cancer. Multiple primary cancers involving lung cancer were also reported. A total of 22,405 cancer patients were identified over a 5-year period, of whom 193 (0.86%) had multiple tumors [9]. Most of the secondary cancers after lung cancer occurred within a 1-year period. There were 3 synchronous (within 2 months), 2 primary malignancies, and 13 metachronous primary cancers. Li et al. reported a 3.4% incidence rate of multiple primary malignancies (185 of 5405) in patients with lung cancer from the Lung Cancer Center of Guangdong, China [10]. The rate of synchronous multiple primary malignancies was 36.6%, according to the definition of a 6-month interval. Digestive system malignancies were the most common (45.7%) involving SPM, and urogenital system malignancies were the third-most-common (18.9%).

An analysis of multiple myeloma reported a cumulative incidence of SPM of 1.33% after a 3-year follow-up [11]. Cheng et al. analyzed 129 digestive-system cancer patients with SPM in a 10-year period from a single medical center in Taiwan and found that gynecologic cancers were the most commonly associated non-digestive SPM. Genitourinary system cancers were the third-most-common. The authors reported that 10 patients were associated with genitourinary tumors and 5 with urinary bladder tumors [12].

Hepatocellular carcinoma (HCC) in Taiwan is very common. Wu et al. analyzed 10 years of data including 45,976 patients with HCC from the National Health Insurance Research Database (NHIRD) [13], of whom 749 (1.6%) were diagnosed with SPM at 90 days after HCC. SPM of the digestive system was the most common and of the genitourinary system was the third-most-common, followed by urinary bladder, renal, and prostate cancers. Another analysis of the NHIRD by Hung et al. reported that ovarian cancer was the most-common cancer [14]. They found 707 SPMs from 12,127 patients (5.8%) over 14 years of follow-up. The standardized incidence rate of bladder cancer was 3.17 with a 95% confidence interval (CI of 1.73~5.32).

## 3. Risk Factors for Multiple Primary Malignancies

Table 1 summarizes the risk factors associated with multiple primary malignancies.

### 3.1. Virus

Viral infection can be a risk factor for second primary cancer. Human papilloma virus (HPV) is well-known to be associated with certain gynecological cancers. Huang et al. recruited 103 patients with oral-cavity squamous cell carcinoma and analyzed their pathological specimens for HPV infection [15]. They found that 30% of the patients were infected with HPV, especially types 16 and 18. These types are highly associated with susceptibility to cancer. Simian virus 40 (SV40) is a potential risk factor for human cancer. SV40 has been reported due to accidental inoculation in humans via vaccine administration since 1960 [16]. Independent from vaccination, SV40 has been reported in ~30% of the human population [17]. Sequences of SV40 DNA have been found in some human cancers such as brain cancer, bone tumors, and malignant mesothelioma [18,19,20]. Although there is no definite conclusion regarding the relationship between SV40 and cancer, a higher prevalence of SV40 antibodies have been detected in oncological patients than normal healthy controls from an indirect ELISA assay [17]. Therefore, SV40 is potentially associated with cancer formation.

About 15% of human cancers may be related to viral infection [21]. The tumorigenesis from viral infection may be slow and inefficient, so that only a few viral infections finally lead to the development of cancer. Viral infection can cause cancer in some patients with immunocompromised status, somatic mutations, genetic predisposition, and exposure to carcinogens [22]. Hepatitis B and C virus are highly associated with hepatocellular carcinoma [23,24]. HPV is associated with many cancers such as cervical cancer, penile cancer, oral cancer, and others [25,26,27]. Other viruses such as Epstein-Barr virus (EBV) and human herpesvirus 8 (HHV-8) are also involved in lymphoma and nasopharyngeal cancer formation [28]. Since viral infections can lead to the development of many cancers, it is possible to develop SPM from a primary malignancy related to viral infection. Therefore, serum viral titers should be checked in these patients.

### 3.2. Chemotherapy

Chemotherapy for primary cancer may be associated with an increased risk of secondary primary cancer. Liu et al. analyzed 4327 patients with newly diagnosed multiple myeloma from the Taiwan National Cancer Registry and NHIRD from 2000 to 2014 [11]. They compared novel-agents (such as bortezomib) and chemotherapy-alone groups for the risk of SPM, and found that the risk of SPM with novel agents was lower than with chemotherapy alone. Therefore, chemotherapy may increase the risk of SPM. Cyclophosphamide (CYP) is commonly used to treat cancers such as non-Hodgkin’s lymphoma, breast, cervical, and pediatric cancers [29]. CYP is also used to treat autoimmune disease or lupus nephritis [30,31]. This issue is beyond this review. However, high-dose CYP therapy has been shown to increase the risk of leukemia, kidney, and bladder cancers [32,33,34].

Cisplatin has been associated with the risk of second cancer. Travis et al. conducted a case-control study on the risk of leukemia in patients with invasive ovarian cancer treated with cisplatin-base chemotherapy in North America and Europe [35] and reported a relative risk (RR) of leukemia of 4.0 (95% CI 1.4~11.4) in platinum-based combination chemotherapy. They found that the risk was dose-dependent when the platinum dosage reached 1000 mg or more, with an RR of 7.6 (*p* for trend <0.001). However, Liang et al. systematically reviewed 28 eligible trials with 7403 patients regarding the risk of second cancer in cancer patients treated with cisplatin [36] and found that compared with non-cisplatin chemotherapy, cisplatin was not significantly associated with the risk of second primary cancer (odds ratio 0.95; 95% CI = 0.67~1.33; *p* = 0.76). Nevertheless, high-dosage platinum-based chemotherapy is still a potential risk for SPM. Elicin et al. retrospectively reviewed the results of 296 head and neck cancer patients, who underwent radiation therapy combined with platinum-based chemotherapy or monoclonal antibody cetuximab therapy [37]. The third group switched from platinum to cetuximab therapy due to toxicity. The results showed that the third group had a significantly higher risk (23.3%) of developing SPM than the former two groups (9.2% and 11.5%, respectively). Therefore, further studies are needed to investigate the risk of platinum compounds for the development of SPM, especially when combined with radiation therapy.

### 3.3. Radiation

Radiation exposure is a potent carcinogen, which is related to dosage and radiation sensitivity. Li et al. analyzed follow-up data from 77,752 Japanese atomic bomb survivors, and found primary cancers in 14,048 patients and SPMs in 1088 patients [38]. There was a linear dose-dependent association between radiation exposure and solid tumors and leukemia. Radiation sensitive organs include the lung, colon, breast, thyroid, and bladder.

Radiotherapy is helpful for certain cancers; however, it may also induce second cancers. The incidence of SPMs following radiotherapy has been estimated to be about 8% in adults within 10 years [39]. In a study of 9807 digestive system cancers associated with 246 SPMs [12], 42.3% of the patients with SPMs had previously received radiotherapy or chemotherapy to treat the first tumor. A study on ovarian cancer associated with SPM by Hung et al. found that radiotherapy had a hazard ratio of 2.07 [14]. In addition to acute radiation exposure and the above reports, de Gonzalez reviewed 28 studies of high-dose exposure (>5 Gy) from radiation therapy including 3434 cancer patients [40] and concluded that there was no linear association between a decrease in the risk and dose-response curve. This discrepancy may be due to improvements in radiation treatment.

### 3.4. Genetics

Early in 1977, Mulvihill et al. investigated multiple primary cancers and found that genetic factors were one of the etiologies [41]. They suggested that family trait should be taken into consideration as a possible cause of multiple primary cancers. Cybuliski et al. reviewed mutated genes associated with cancer formation, and found that features of hereditary cancer included a positive family history, early age at onset, and multiple primary tumors. They suggested that genetic testing is indicated for such patients [42]. The NCCN guidelines also state that indications for high-penetrance breast cancer and/or ovarian cancer include susceptibility genes such as BRCA1, BRCA2, PTEN, TP53, CDH1, and others [43].

Lynch syndrome is a mutation of DNA repair genes including *MLH1*, *MSH2*, *MSH6*, *PMS2*, and *EPCAM*, which can cause many cancers at a young age. These genes work together to repair DNA errors during cell division, and mutations in these genes can cause failure to repair DNA errors and subsequently cancer formation. The most well-known cancer associated with Lynch syndrome is hereditary nonpolyposis colorectal cancer (HNPCC), and Lynch syndrome also has increased risks of gastrointestinal, liver, kidney, brain, and skin cancers [44]. Another well-known genetic mutation involving multiple tumors is the BRCA gene [45]. Besides breast and ovarian cancers, mutations of BRCA1 and BRCA2 genes also increase the risk of pancreatic and prostate cancers. Multiple endocrine neoplasia (MEN1 and MEN2) is also involved in many endocrine neoplasia and thyroid cancers, in which genetic mutations have been reported [46,47,48]. In addition to the above genetic disorders, there are several genetic mutations associated with breast and ovarian cancers such as Li-Fraumeni syndrome, hereditary diffuse gastric cancer syndrome, Peutz-Jeghers syndrome, and PTEN Hamartoma tumor syndrome [49]. Hereditary cancer diseases are associated with many genetic mutations, however, further investigations are needed to clarify which mutations are involved. Smoking is a common risk factor for many cancers. Liu et al. analyzed patients with lung cancer from a database of Taipei Veterans General Hospital over a 5-year period, and found that smoking was a significant risk factor for lung cancer and multiple primary malignancies [9]. However, Liu et al. reported that 68.0% of the patients with multiple primary malignancies involving lung cancer had never smoked [10]. The powerful association between smoking and developing SPM may be related to the continuous habit of smoking and alcohol consumption [50]. Stopping smoking is very important for cancer survival and to prevent SPMs. Tabuchi et al. retrospectively analyzed 29,795 primary cancer patients regarding smoking and SPM [51] and found that smoking was highly associated with primary cancer and SPM. Smokers who have recently quit have increased cancer survival than those who continue. Smoking increases the risk of oral/pharyngeal, lung, esophageal, stomach, and hematologic SPMs.

### 3.5. Associated Cancer

Some cancer characteristics may by highly association with the same kind of malignancy. Patients with multiple myeloma have an extremely high risk of hematologic malignancy. Tzeng et al. analyzed a total of 3970 patients with multiple myeloma from 1997 to 2009 through the National Cancer Registry of Taiwan, with a focus on secondary hematologic malignancies [52]. They found an 11-fold higher incidence of second hematologic malignancies in the multiple myeloma group than in the comparison cohort without multiple myeloma (adjusted hazard ratio 13.0; 95% CI 7.79~21.6). The incidence of myeloid leukemia was also much higher than in the comparison cohort) (RR 21.2 vs. 1.36, respectively). The risk was especially high in young patients. Therefore, there might be a link between tumors such as sharing common risk factors or genetic linkage.

### 3.6. Environment

Environmental factors are also a risk for developing SPM. It is generally accepted that environment–gene interactions cause cancer. Environmental factors associated with cancers include smoking, alcohol, organic and inorganic chemicals, sunlight and ionizing radiation, diet and obesity, hormonal therapy, and air and water pollution [53]. Genes associated with cancers included four groups: oncogenes, tumor suppressor genes, DNA repair genes, and apoptosis-related genes. Up to 93% of all human cancers have been reported to be caused by environment–gene interactions [54]. However, Rudolph et al. reviewed the available evidence on environment–gene interactions and concluded that there was only a weak association for breast cancer [55]. Patients who are susceptible to cancers affected by environmental factors should have predisposing genetic traits [56]. Therefore, there is currently no strong evidence of an association between environmental factors alone and the pathogenesis of SPM. Further studies on environmental factors and susceptible genetic traits are necessary.

### 3.7. Betel Quid Chewing

Betel quid chewing is a well-known risk factor for oral cancer in many Asian countries including Myanmar, India, and Taiwan [57,58,59]. Besides oral cancer, betel quid chewing is also associated with pharynx cancer [60]. Betel quid chewing and smoking are also associated with increased risks of oral, esophageal, lung, pancreatic, and liver cancers in Taiwan [61]. Adel et al. performed a long-term follow-up study of oral cancer patients in a betel quid chewing endemic area with secondary to fourth primary cancers [62]. They reported that the incidence rates of second, third, and fourth primary tumors at 5 and 10 years were 20.2%/34.6%, 4.0%/8.6%, and 1.0%/2.3%, respectively. Overall survival was significantly decreased with multiple primary tumors associated with oral cancer. Betel quid contains arecoline and arecaidine, which play an important role in the pathogenesis of oral cancer [63]. These alkaloids may dysregulate oral keratinocytes during mitosis, promote instability of the genome, and induce inflammation [64,65]. Therefore, betel quid chewing plus smoking are risk factors for multiple primary tumors. Early quitting of betel quid chewing maybe the best way to prevent SPMs.

### 3.8. Chronic Kidney Disease

Chronic kidney disease may be associated with several cancers via direct or indirect routes [66]. Nephrotoxic agents such as analgesic can cause nephropathy and kidney cancer [67,68,69]. The indirect relationship between chronic kidney disease and cancer may be related to dialysis or transplantation status of immunosuppression or immune dysfunction [70]. Underlying kidney diseases such as acquired adult polycystic kidney disease have been associated with renal cell carcinoma by increasing DNA damage and repair and decreasing antioxidant capacity [70].

## 4. Incidence of Second Primary Tumors in the Genitourinary Tract

Second primary tumors in the genitourinary tract have also been frequently reported [71]. In a 10-year follow-up study of prostate cancer in Victoria, Australia, one in nine were secondary primary cancers. Common second tumors were found in the colon, rectum, lung, melanoma, and bladder. In kidney cancer, secondary tumors were also found in one in nine of the male patients. Colon and lung cancers were the most-common. In addition, renal pelvis cancer was associated with a high cumulative risk of second primary cancer in men (one in six). Lung, pancreas, melanoma, and bladder cancers were common, and all were strongly associated with smoking [71]. These data were reported from Australia, with a comment that renal pelvis cancer was less frequently seen. In, Taiwan, renal pelvis cancer is not uncommon and accounts for over 50% of all kidney cancers [72,73]. The cumulative risk may be underestimated. Another issue that remains to be confirmed is whether bladder cancer is a second primary cancer or a recurrent cancer.

## 5. Early Detection of Second Primary Malignancy

SPM may develop in a small population of primary cancers. However, synchronous secondary or even third primary malignancies increase the difficulty of cancer treatment and decrease patient survival. Identifying the occurrence of SPM early in the same patient would allow for an earlier diagnosis and treatment, which could improve follow-up care and obtain good survival, so it is an important issue [74].

Early detection often occurs incidentally when computed tomography (CT) is performed to stage a patient with a known malignancy, and it is usually noted first by a radiologist when a new lesion atypical for metastases is detected. The most frequent incidentally detected second tumor is hepatocellular carcinoma, followed by renal carcinoma, lung carcinoma, and bladder carcinoma [75].

Identifying the aforementioned risk factors for SPMs including viral infection, radiotherapy, chemotherapy, smoking, betel quid chewing, genetics, specific environmental factors, and/or cancer-promoting lifestyle aspects would also be helpful [2]. Carefully surveying primary cancer patients with many associated risk factors for SPM may facilitate the early detection of SPM.

## 6. Conclusions

Several risk factors are associated with cancer formation and SPMs. Performing risk factors tests may be helpful for some patients with primary cancer, for the early detection of secondary cancer. The reported incidence of SPMs from first primary malignancy ranges from 1.3% to 5.8%. Urothelial cell carcinoma is prone to local recurrence, distal recurrence, urinary bladder recurrence from upper tract primary malignancy, and contralateral occurrence.

## Figures and Tables

**Table 1 diagnostics-12-01940-t001:** Risk factors associated with multiple primary tumors.

Figure	Category	Organs of Study Cancer	Secondary, Third, and Fourth Primary Tumors	Prevention or Tests
Virus	HPV SV40 Epstein-Barr virus and human herpesvirus 8 Hepatitis B and C virus	Gynecological cancer Brain, bone Nasopharyngeal carcinoma Hepatocellular carcinoma	Penile cancer, oral cancer Malignant Mesothelioma, Lymphoma	HPV test in women PCR or ELISA for SV40 HBV and HCV titer
Chemotherapy Radiation Genetics Smoking	Cyclophosphamide Cisplatin BRCA1, BRCA2, PTEN, TP53, CDH1 Lynch syndrome (MLH1, MSH2, MSH6, PMS2 or EPCAM genes) BRCA1, BRCA2 MEN1, and MEN2	Non-Hodgkin’s lymphoma, breast, cervical and pediatric cancers ovarian cancer, and oral cancer Digestive cancers Breast Colon Breast and ovary Endocrine, and thyroid Lung	Leukemia, kidney, and bladder cancer Leukemia Gynecological, genitourinary, lung, and breast cancers Ovarian cancer Gastrointestinal, liver, kidney, brain, and skin cancers Pancreas, prostate adrenal cortical tumors, carcinoid tumors, facial angiofibromas, collagenomas, and lipomas Another lung cancer, upper gastrointestinal, genitourinary, and colorectal cancers	Genetic test Cessation smoking
Cancer association	Same characteristics	Multiple myeloma	Hematologic malignancy	
Betel quid	Arecoline and arecaidine alkaloids	Oral cancer	Pharynx, esophageal, lung, pancreatic, and liver cancers	Quit betel quid chewing

## Data Availability

Not applicable.

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
