# Peer review of "Synchronous/Metachronous Multiple Primary Malignancies: Review of Associated Risk Factors"

_diagnostics, 2022, doi:10.3390/diagnostics12081940_

Round 1
Reviewer 1 Report
Review of scientific report entitled “Synchronous/Metachronous Multiple Primary Malignancies: Review of Associated Risk Factors” by Szu-Ying Pan et al.
The authors aimed to explore the incidence of secondary primary malignancy (SPM) in Taiwan with a focus on genitourinary malignancies. Several risk factors are related to the occurrence of SPM, including viral infection chemotherapy, radiation, genetics, smoking, betel quid chewing and environmental factors. Early survey of SPM is indicated in first primary malignancy patients with these associated factors. In this study, the authors summarize several risk factors related to the occurrence of SPMs, which may help in their early detection and consequently better survival. In addition, genitourinary malignancies have been reported to be the third most common SPM, but they have not been reported as first primary cancers. In this review, the authors comprehensively discuss genitourinary malignancy-related SPM.
Although the review is interesting, unfortunately there are some major criticisms:
1. The review often appears confusing, and difficult to understand. Recent papers have analyzed the incidence of Synchronous / Metachronous Multiple Primary Malignancies, such as the review entitled "Risks and cancer associations of metachronous and synchronous multiple primary cancers: a 25-year retrospective study" by Tanjak P. et al. (BMC Cancer volume 21, Article number: 1045 (2021). Rather than analyzing the incidence of this type of multiple neoplasia in the Taiwanese population which deserves a separate paragraph, the review of the pathogenetic factors involved in development is much more interesting. In this sense, the various factors should be analyzed more extensively reporting those tests or surveillance strategies that can reduce the incidence to develop multiple primary malignancies by preventing it where possible and/or recognizing it at an early stage. For example, regarding the viral factors could be useful to known if a specific virus can be detected in the development of a certain tumor and then to set a strict follow-up. The table should be completely revised and in the light of what is suggested, adding a new column in which the various and possible preventive tests are reported.
2. The part of the review on multiple urothelial tumors should be completely eliminated as nothing new is added in this area that is not already known.
3. Figure 1 is not necessary.
Author Response
Comments and Suggestions for Authors
The authors aimed to explore the incidence of secondary primary malignancy (SPM) in Taiwan with a focus on genitourinary malignancies. Several risk factors are related to the occurrence of SPM, including viral infection chemotherapy, radiation, genetics, smoking, betel quid chewing and environmental factors. Early survey of SPM is indicated in first primary malignancy patients with these associated factors. In this study, the authors summarize several risk factors related to the occurrence of SPMs, which may help in their early detection and consequently better survival. In addition, genitourinary malignancies have been reported to be the third most common SPM, but they have not been reported as first primary cancers. In this review, the authors comprehensively discuss genitourinary malignancy-related SPM.
Although the review is interesting, unfortunately there are some major criticisms:
- The review often appears confusing, and difficult to understand. Recent papers have analyzed the incidence of Synchronous / Metachronous Multiple Primary Malignancies, such as the review entitled "Risks and cancer associations of metachronous and synchronous multiple primary cancers: a 25- year retrospective study" by Tanjak P. et al. (BMC Cancer volume 21, Article number: 1045 (2021). Rather than analyzing the incidence of this type of multiple neoplasia in the Taiwanese population which deserves a separate paragraph, the review of the pathogenetic factors involved in development is much more interesting. In this sense, the various factors should be analyzed more extensively reporting those tests or surveillance strategies that can reduce the incidence to develop multiple primary malignancies by preventing it where possible and/or recognizing it at an early stage. For example, regarding the viral factors could be useful to known if a specific virus can be detected in the development of a certain tumor and then to set a strict follow-up. The table should be completely revised and in the light of what is suggested, adding a new column in which the various and possible preventive tests are reported.
- The part of the review on multiple urothelial tumors should be completely eliminated as nothing new is added in this area that is not already known.
- Figure 1 is not necessary.
Response:
Thanks for your positive comments, we have revised the manuscript according to your and the other reviewers’ valuable suggestions. Your understanding and further editorial consideration will be very much appreciated.
1.Yes, we have revised our manuscript regarding “incidence” in the text according to the report of Tanjak et al. We also have added a new column regarding lists of preventive tests in the Table 1.
2.We removed the review part on multiple urothelial tumors.
3.Yes, we removed Figure 1 in the revised manuscript.
Reference:
Tanjak P, Suktitipat B, Vorasan N, Juengwiwattanakitti P, Thiengtrong B, Songjang C, Therasakvichya S, Laiteerapong S, Chinswangwatanakul V. Risks and cancer associations of metachronous and synchronous multiple primary cancers: a 25-year retrospective study. BMC Cancer. 2021 Sep 23;21(1):1045.
Reviewer 2 Report
The review is interesting because it puts together relevant information about synchronous and metachronous multiple primary malignancies. I do not understand why genito-urinary malignancies deserve a specific heading in the review. I would suggest to immerse the information provided in this paragraphs in the body of the manuscript.
Author Response
Comments and Suggestions for Authors
The review is interesting because it puts together relevant information about synchronous and metachronous multiple primary malignancies. I do not understand why genito-urinary malignancies deserve a specific heading in the review. I would suggest to immerse the information provided in this paragraphs in the body of the manuscript.
Response:
Yes, we removed this part of review. Thanks.
Round 2
Reviewer 1 Report
The new version of the review is improved but the authors should revise the abstract that takes into account the changes made. Moreover, the preventive tests and the use of these should be reported in the specific paragraph and in the conclusion part.
Author Response
Comments and Suggestions for Authors
The new version of the review is improved but the authors should revise the abstract that takes into account the changes made. Moreover, the preventive tests and the use of these should be reported in the specific paragraph and in the conclusion part.
Response:
Thanks for your positive comments, we have second revised the manuscript according to your and the other reviewers’ valuable suggestions. Your understanding and further editorial consideration will be very much appreciated.
1.Yes, we have rewritten our manuscript regarding “abstract” and “discussion” according to the report.
2.We also add the prevention tests in the text such as quit chewing betel quid and perform viral titers check. In addition, some prevention methods have been written in the original manuscript such as SV40 ELISA test, stopping smoking and genetic tests.
Thanks.